# OpenReview forum: "Aligning to Thousands of Preferences via System Message Generalization"
_NeurIPS.cc/2024/Conference — NeurIPS 2024 poster_

### Official Review · Reviewer_bEUV · 2024-06-30

**Soundness:** 2
**Presentation:** 2
**Contribution:** 3
**Rating:** 6
**Confidence:** 5

**Summary:**

In this paper, the authors introduce a novel method to align LLMs with diverse user preferences without requiring continual retraining for each specific preference. The approach utilizes a unique system message protocol that guides LLMs to produce responses tailored to specific, nuanced user preferences.

**Strengths:**

1. The idea is novel: adapting to different users' preferences by training on preference data with different system prompts.
2. It is intuitively a useful research, effectively strengthening the alignment effectiveness by clarifying the user identities besides the alignment data themselves.
3. The paper is well-written and easy to follow.

**Weaknesses:**

1. How many humans did you hired to perform the human evaluation? There lacks a illustration on this. If we want to illustrate its ability to cater to a wide range of peoples with different backgrounds, then we should also apply a wide range of people to evaluate.
2. I feel that there lacks an illustration to show that the model is not becoming verbose. As all of the benchmarks you use prefer verbose responses.

**Questions:**

Is this a way to "hack" the Chatbot Arena? (in the right way lol)

**Limitations:**

Yes

---

> ### Author Rebuttal · Authors · 2024-08-07
>
> Dear reviewer bEUV, we deeply appreciate your valuable feedback.
>
> We provide responses to your comments in the weaknesses (W) and questions (Q) section.
>
> ---
>
> **Details of human annotation/evaluation (W1)**
>
> We employ 14 undergraduate students aged 20-24 from various majors, consisting of 6 males and 8 females. Hiring annotators with diverse backgrounds and preferences is an important aspect of human evaluation. However, in this study, human annotators assess the quality of the {system message, user prompt, response}  set according to clear criteria, regardless of their personal preferences. They also evaluate which model better reflects a given preference in response generation. The Stage 1 and Stage 2 human evaluation process  is detailed in Appendix F.1. Therefore, while individual preferences of the annotators are valuable, they do not apply in our specific context.
>
> ---
>
> **Analysis of response verbosity (W2)**
>
> In Appendix G.1 Figure 8, we plot the length distribution of responses and reference answers on Multifaceted Bench. It indeed shows that Janus’s responses are longer than other LLMs including Mistral 7B Instruct v0.2 and GPT 3.5 Turbo. Since the distribution of Janus responses and that of reference answers made by GPT-4-Turbo-0125 are similar, it can be seen as the result of supervised learning. Still, Table 3 shows that Janus outperforms models including the same Mistral 7B Instruct v0.2 and GPT 3.5 Turbo on AlpacaEval 2.0 when compared using the *length-controlled* (LC) win rate. The LC win rates are a debiased version of the win rates that control for the length of the outputs and are reported to improve leaderboard correlation [1]. Since Janus exhibits high performance consistently across benchmarks including length-controlled measures, we stress that verbosity does not play as a critical factor for evaluators to prefer Janus responses over those from other models.
>
> ---
>
> **Hacking the Chatbot Arena (Q1)**
>
> Could you please elaborate on your question regarding our method 'hacking' the Chatbot Arena? Are you suggesting that this hacking is a result of biases in the datasets or models, or are you referring to biases in the evaluation methods?
>
> ---
>
> [1] Dubois et al.. Length-Controlled AlpacaEval: A Simple Way to Debias Automatic Evaluators. arXiv 2024.

---

> > ### Comment · Reviewer_bEUV · 2024-08-10
> >
> > Hi authors,
> >
> > Thanks for your responses!
> >
> > To clarify the question: can this method be used as a way to improve the models' ranking on chatbot arena simply by fitting the human preferences or arena users' preferences?

---

> ### Author Response · Authors · 2024-08-12
>
> Dear reviewer bEUV, thank you for the clarification!
>
> Our training dataset and method does improve the model in preference-based benchmarks like AlpacaEval 2.0 (Table 3).
>
> However, our method is not centered on how to extract user preferences from the human population. Instead, we created training data simulating user preferences by distilling GPT-4 Turbo’s knowledge, which itself was effective in achieving both personalization and general helpfulness.
>
> Also, since Chatbot Arena is updated via *real-time user votes*, we do not see it feasible to fit the arena users’ preferences as well.
>
> We emphasize that our method cannot be used to mine or fit existing users’ preferences. The intended use of our method is to align to diverse individuals’ preferences and generalize to unseen system messages.
>
> We hope this answers your question!

---

### Official Review · Reviewer_k2dN · 2024-07-05

**Soundness:** 3
**Presentation:** 3
**Contribution:** 2
**Rating:** 5
**Confidence:** 4

**Summary:**

This paper addresses the issue that humans inherently have diverse values, while current LLMs are primarily aligned with general public preferences such as helpfulness and harmlessness. Previous work has trained new reward models (RMs) and LLMs for individual preferences, which is time-consuming and costly.

The authors propose a new paradigm where users specify their values within system messages. Specifically, they create the Multifaceted Collection, containing 65k user instructions, each with 3 system messages, resulting in 192k combinations of fine-grained values with the assistance of GPT-4. They then train mistra-7b-v0.2 into Janus-7B and test it on 921 prompts collected from 5 benchmarks. Compared to leading open-sourced and proprietary models using pairwise ranking human evaluation or GPT-4-as-a-judge, Janus-7B achieves a higher win rate while maintaining harmlessness.

**Strengths:**

The paper proposes a new paradigm to address the problem of aligning to fine-grained values without repeatedly training multiple RMs and LLMs. It also creates a large-scale preference dataset with fine-grained system prompts representing multiple alignment targets. Detailed experiments across many leading open-source and proprietary models validate the method's effectiveness.

**Weaknesses:**

While the paper is well-written and presents a solid analysis, several weaknesses need addressing:

1. The construction of the Multifaceted Collection is costly and not scalable. GPT-4 generates preference sets for each of the 65k instructions, converts each of the 198k preference sets into a system message, and crafts gold-standard multifaceted responses for each system message.
2. The role of system message generalization has not been clarified. What are the differences between varying the system messages for individualized alignment targets and simply using different instructions in the user prompt while maintaining the system message unchanged to reach the specified alignment targets?

**Questions:**

1. What are the differences between varying the system messages for individualized alignment targets and simply using different instructions in the user prompt while maintaining the system message unchanged to reach the specified alignment targets? It seems that I can just use specialized instructions in user prompts to reach the same goal of aligning to individualized values. Can you comment on this?
2. There are several typos:
   - Line 136: 5 datasets, not 4
   - Line 152: Appendix B.1?
   - Line 286: Appendix G.1? And so on

**Limitations:**

The construction of the Multifaceted Collection is costly and not scalable. And the role of system message generalization is unclear. It is not explained how varying system messages for individualized alignment targets differs from using different instructions in user prompts while keeping the system message unchanged.

---

> ### Author Rebuttal · Authors · 2024-08-07
>
> Dear reviewer k2dN, thank you for the important remarks that will help us better shape our contributions.
> We will address your concerns in weaknesses (W1, W2) and questions (Q1, Q2) below.
>
> ---
>
> **Redefining the scalability of our approach (W1)**
>
> Creating a synthetic dataset can indeed be costly, and it is challenging to consider the values of every individual in the world when generating data. However, this does not necessarily mean that the process is not scalable. According to previous studies [1, 2, 3], training with a sufficiently diverse and large dataset can enable language models to address problems requiring skills they have not explicitly learned. As shown in Table 2, all the preferences used in the benchmark are ones that Janus has not encountered during training. Nonetheless, Janus's superior performance compared to the baselines suggests that it possesses significant generalizability. We believe that our approach can overcome the scalability problem in individualized alignment by previous personalized RLHF approaches (Section 2).
>
> ---
>
> **Discussing the role of system messages for individualized alignment (W2, Q1)**
>
> The core of our methodology is to verbalize preferences in the model input and train a model on many of such instances to improve generalization. Our method is an extension of instruction tuning, where we *control preferences in the instruction* to give stronger signals to the model on fine-grained preferences than on general helpfulness and harmlessness dimensions. In this aspect, your comment on using different instructions in the *user prompt* to achieve individualized alignment is valid. Applying our training recipe using user prompts instead of system prompts would yield similarly strong results.
>
> Nevertheless, we tried to separate the preference-specific instructions in the system message, since there is much underexplored potential in it for reaching alignment targets. From an application standpoint, the best scenario would be when the user specifies their preferences by themself in the user prompt, but in reality, users would not take much effort to do so. A way to improve user experience would be the developers  inferring preferences (possibly based on past conversation history), verbalizing them in the system message, and aligning the model response with the user’s *hidden* interests. A model can be trained to assign highest privilege to the system message for more granular preference steering [4]. However, there is little work that experiments with diverse system messages (Section 2), so we aimed to conduct a study on the benefits of varying system messages for the same user prompt.
>
> ---
>
> **Typos (Q2)**
> Thank you for the suggestions. We will go over the draft carefully and revise any inaccuracies.
>
> ---
>
> [1] Sanh et al.. Multitask Prompted Training Enables Zero-Shot Task Generalization. ICLR 2022.
>
> [2] Longpre et al.. The Flan Collection: Designing Data and Methods for Effective Instruction Tuning. ICML 2023.
>
> [3] Kim et al.. Prometheus: Inducing Fine-Grained Evaluation Capability in Language Models. ICLR 2024.
>
> [4] Wallace et al.. The Instruction Hierarchy: Training LLMs to Prioritize Privileged Instructions. arXiv 2024.

---

> > ### Comment · Reviewer_k2dN · 2024-08-09
> >
> > Thank you for your clear response. The role of system messages in individualized alignment can indeed be seen as a method to infer users' preferences. However, in practice, what specific system message would you use to better align with users' goals? Would it be based on past conversation history? If so, have you experimented with "developers inferring preferences (possibly based on past conversation history), verbalizing them in the system message"? Additionally, can system messages inferred from past conversation history be generalized using JANUS? My main concern is that aligning with varying preferences through system message generalization seems somewhat similar to using diversified instructions in instruction tuning, aiming to enable generalization to other instructions.

---

> ### Author Response · Authors · 2024-08-12
> **Author Response to Reviewer k2dN (1/2)**
>
> Dear reviewer k2dN, thank you for raising the discussion. We took your concerns carefully and will provide detailed comments below.
>
> First, as per your main concern on our method’s similarity with instruction tuning, we agree with the following statement:
>
> > aligning with varying preferences through system message generalization seems somewhat similar to using diversified instructions in instruction tuning, aiming to enable generalization to other instructions.
>
> System message itself is an instruction that the model should follow, so the basis of system message generalization is indeed in instruction tuning.
> With sufficiently many diversified system messages, our method takes advantage of the instruction tuning recipe to generalize to unseen system messages (i.e., individualized alignment targets).
>
> However, our focus on the *content* differs from previous instruction tuning works. While previous works scale inputs, input-free tasks, or tasks per input (three paradigms described in [1]) in instructions, **we aim to scale *meta-instructions*, instructions that guide how to respond to subsequent instructions**. Our motivation was that different people expect different responses on the same instruction, so such meta-instructions that set preferences for task execution should also be taught to the model. Including this in instruction tuning helps the model approach various types of prompts and dynamically adjust response strategies, as shown in Section 5.1 and 6.2. Method-wise, our hierarchical data generation strategy enables careful curation of those meta-instructions to specifically simulate user preferences instead of presenting irrelevant challenges or hinting solutions with respect to the instruction. To the best of our knowledge, **there is no work that shares a similar motivation with us or has publicly released a training dataset containing meta-instructions**.
>
> ---
>
> [1] Lou et al.. MUFFIN: Curating Multi-Faceted Instructions for Improving Instruction Following. ICLR 2024.

---

> ### Author Response · Authors · 2024-08-12
> **Author Response to Reviewer k2dN (2/2)**
>
> Lastly, we would like to elaborate on why we have set meta-instructions in the system message and how to further utilize them. While some meta-instructions can only be defined by a specific user message, it can also be more general. For example, the following system message in our dataset is associated with a user prompt giving a specific hyperbola equation, but is also applicable to other kinds of problems.
>
> ```
> You are a mathematics coach, specializing in presenting complex mathematical concepts to users with an intermediate understanding. Your teaching method is known for its succinctness, offering clear, direct explanations and step-by-step walkthroughs without overcomplicating the information. To facilitate learning, incorporate visual aids like charts and diagrams, making abstract concepts tangible and easier to understand. Your approach always considers the users' math anxiety, striving to create an engaging, supportive environment that reduces stress and fosters confidence. Your aim is to make math accessible and less intimidating, ensuring users can grasp the essentials of mathematical theories quickly and effectively, with patience and creativity at the core of every interaction.
> ```
>
> In this sense, **meta-instructions can be effective when set programmatically on relevant user messages in the form of system messages**. This (i) allows reflecting the users’ common needs on various instructions and (ii) lowers user burden. In practice, OpenAI has already shipped similar features like custom instructions [2] and memory controls [3] in ChatGPT; a user can specify their preferences in the custom instruction by oneself (e.g., *When I ask you for code, please just give me the code without any explanation on how it works.*) or ChatGPT will memorize the details itself from the conversations. When user preferences are gathered or summarized from interactions, it can be decided to be included as part of the system message without user specification for seamless chat experience.
>
> Exactly how to infer user preferences or how to manage user controls on system messages is application-dependent and beyond the scope of our work (noted in broader impact in Appendix I). In the line of works on dialogue summarization [4, 5], we look forward to future work to explore the challenges of inferring preferences from conversations. The focus of our work lies more in **representing user preferences in the form of meta-instructions** and **training models to follow meta-instructions**. Also, from a technical perspective, training meta-instructions in system messages with accompanying gold responses would make the model (with appropriate system tokens) better distinguish between meta-instructions and instructions and tailor responses to the higher-level guidance.
>
> We greatly value your feedback and will improve our writing to shape the storyline better.
>
> ---
>
> [2] OpenAI. Custom instructions for ChatGPT. OpenAI Blog 2023.
>
> [3] OpenAI. Memory and new controls for ChatGPT. OpenAI Blog 2024.
>
> [4] Zou et al.. Unsupervised Summarization for Chat Logs with Topic-Oriented Ranking and Context-Aware Auto-Encoders. AAAI 2021.
>
> [5] Zhao et al.. TODSum: Task-Oriented Dialogue Summarization with State Tracking. arXiv 2021.

---

> > ### Comment · Reviewer_k2dN · 2024-08-12
> >
> > Thanks for your detailed response. Your responses solve my concern to some extent. I maintain my score.

---

### Official Review · Reviewer_uZmQ · 2024-07-09

**Soundness:** 4
**Presentation:** 3
**Contribution:** 3
**Rating:** 6
**Confidence:** 4

**Summary:**

This paper aims to align Large Language Models (LLMs) with individual user preferences at scale. The authors propose a paradigm where users specify their values within system messages to guide the LLM's generation behavior. To address the challenge of generalizing to diverse system messages, the authors create the MULTIFACETED COLLECTION, a dataset with 192k unique combinations of values and instructions. They train a 7B LLM called JANUS using this dataset and demonstrate its effectiveness in adhering to various user preferences without the need for retraining for each individual. The JANUS model outperforms other models like Mistral 7B Instruct v0.2, GPT-3.5 Turbo, and GPT-4 in benchmarks that assess response helpfulness.

**Strengths:**

1. The creation of the MULTIFACETED COLLECTION, as well as the trained models, makes a good contribution to the community for studying diverse preference alignment.
2. The idea of utilizing system messages, while simple, is demonstrated as a scalable solution to the problem of individualized LLM alignment.
3. The paper shows that training with a diverse array of system messages not only supports personalized responses but also enhances alignment with general public preferences.

**Weaknesses:**

While widely adopted, the GPT4-based data synthesis process may introduce some artifacts. The authors should include quantitative and qualitative analyses to investigate the potential biases and representativeness in human preferences in the collected MULTIFACETED COLLECTION.

**Questions:**

How do you measure the diversity of user preferences in MULTIFACETED COLLECTION?

**Limitations:**

The authors have discussed the limitations.

---

> ### Author Rebuttal · Authors · 2024-08-07
>
> Dear reviewer uZmQ, we appreciate your thoughtful comments.
>
> Our responses to the issues raised in the weaknesses (W1) and questions (Q1) sections are detailed in the global responses (G2: Diversity, G4: Bias).
>
> ---
>
> **Diversity of User Preferences (Q1)**
>
> We measured the ROUGE-L score among user preferences in the training data and found an average similarity of about 0.2. Compared to previous works, this indicates sufficient diversity in the preferences present in our dataset. Furthermore, the model trained with this data, Janus, generates more diverse responses compared to baselines. This demonstrates the diversity of our dataset.
>
> ---
>
> **Bias and representativeness of human preferences (W1)**
>
> We tested Janus’ performance on three social bias benchmarks and discovered that Janus exhibits a bias level comparable to LLaMA 3 8B Instruct across all tasks and surpasses Mistral 7B Instruct v0.2 in most tasks. No significant bias issues are observed in Janus relative to other models.
>
> We also followed experiments in previous work to measure the similarity between the human distribution and model distribution on survey questions and personality tests. Results using the Jenson-Shannon distance show that Janus shows 2x smaller decrease in similarity to Mistral 7B Instruct v0.2. The similarity is especially evident in terms of entropy distributions. These findings suggest that Janus is reasonably calibrated to the human population and that Multifaceted Collection can be representative of diverse human preferences to a large degree.
>
> Please see more information in G2 and G4 of our global response.

---

> > ### Comment · Reviewer_uZmQ · 2024-08-08
> >
> > Thanks for your response. I have read the author response, and I maintain my rating.

---

### Official Review · Reviewer_eXeU · 2024-07-17

**Soundness:** 3
**Presentation:** 3
**Contribution:** 3
**Rating:** 5
**Confidence:** 4

**Summary:**

This paper introduces JANUS, a novel approach to aligning large language models (LLMs) with diverse individual user preferences without retraining. The key contributions are the MULTIFACETED COLLECTION: A dataset of 192k diverse system messages reflecting varied user preferences, paired with 65k instructions. JANUS: A 7B parameter LLM trained on this dataset to generalize to unseen system messages at test time. Evaluation showing JANUS outperforms several larger models, including GPT-3.5 and GPT-4, in adhering to specified preferences.

**Strengths:**

1. Using diverse system messages for alignment is novel and creative. It tackles the challenge of personalized alignment in a scalable way, avoiding the need to retrain models for each user.
2. The methodology is rigorous, with careful dataset construction and comprehensive evaluations against strong baselines. The use of human evaluation alongside automated metrics strengthens the results.
3. This paper is well-structured and clearly written. The approach and results are explained thoroughly, with helpful figures illustrating key concepts. It work has potentially high impact, addressing a crucial challenge in AI alignment. The ability to adapt to individual preferences without retraining could be transformative for deploying personalized AI systems at scale.

**Weaknesses:**

1. The paper doesn't deeply examine potential risks of allowing such flexible preference specification, such as potential for misuse or unintended biases. - There's no discussion on how to handle conflicting or ethically questionable preferences.
2. While the overall approach is effective, it's unclear which components contribute most to the performance gains.
For example, how much does the hierarchical nature of the preference augmentation contribute versus simply having a large number of diverse preferences? The impact of the number of unique system messages or instructions is not explored.
3. The study focuses on 7B parameter models, primarily comparing to other 7B models or larger. It's unclear how well the approach scales to smaller models or much larger models (e.g., 70B+). The comparison to LLaMA 3 8B Instruct is promising, but more exploration of scale effects would be valuable.

**Questions:**

1. Have you explored any potential negative consequences of allowing such flexible specification of preferences through system messages? Are there safeguards to prevent misuse or handling of conflicting ethical preferences?
2. How does the performance of JANUS change as you vary the number of unique system messages in the training data? Is there a point of diminishing returns, and how does this relate to the hierarchical structure of your preference augmentation?
3. Have you tested the approach on models significantly larger or smaller than 7B parameters? How does the effectiveness of this method scale with model size, especially compared to the scaling behavior of traditional fine-tuning approaches?
4. How consistent is JANUS in maintaining the specified preferences across a long conversation or multiple diverse tasks? Have you evaluated this aspect, and if so, what metrics did you use?
5. The paper mentions outperforming larger models like GPT-3.5 and GPT-4. Have you analyzed why your approach seems to be more effective than simply scaling up model size for this task? Could this indicate a fundamental advantage of your method over scale alone?
How did you ensure the quality and diversity of the generated system messages and reference answers in the MULTIFACETED COLLECTION? Did you implement any specific measures to mitigate potential biases introduced by using GPT-4-Turbo-0125 for generation?

**Limitations:**

None.

---

> ### Author Rebuttal · Authors · 2024-08-07
>
> Dear reviewer eXeU, we deeply appreciate your constructive feedback.
> Here we address the weaknesses (W) and questions (Q) that you raised on our work.
>
> ---
>
> **Discussing the potential risks of allowing flexible preference specification and available safeguards (W1,Q1)**
>
> We agree that training a model with flexible preference specifications may increase its vulnerability to misuse, such as jailbreaking. To address this, we incorporated safety considerations into our data generation process by including a harmlessness dimension in every system message. As discussed in G3, our method does not significantly compromise safety, as demonstrated through evaluations of both the dataset's safety and the model’s likelihood of generating toxic responses to harmful prompts.
> In future work, we plan to enhance our approach by filtering toxic data from the dataset, increasing the proportion of safety-related data in the training set, and utilizing AI moderation technologies like LLaMA-Guard and Prompt-Guard.
>
> ---
>
> **Demystifying the factors that contribute to performance gains (W2, Q2, Q5)**
>
> **(1) Number of training data**
>
> We conducted an ablation study to examine the effects of data scaling. Janus 7B was trained on a dataset of 197k instances, where each user prompt is paired with 3 different system messages and responses. To test the effects of scaling, we reduced the number of system messages and responses per user prompt to 2 and 1, resulting in datasets of 132k and 66k instances, respectively, and trained separate models. The results indicate that, across five benchmarks, the scaling effect is evident: increasing the amount of training data leads to higher scores.
>
> | \# system message per instruction | \# total train instances | mf -AlpacaEval | mf-FLASK | mf-Koala | mf-MT-Bench | mf-Self-Instruct |
> | --------------------------------- | ------------------------ | -------------- | -------- | -------- | ----------- | ---------------- |
> | 1                       | 66k                      | 4.415          | 4.017    | 4.358    | 4.083       | 4.025            |
> | 2                    | 132k                     | 4.427          | 4.052    | 4.39     | 4.1         | 4.01             |
> | 3                     | 197k (→ Janus)                    | 4.43           | 4.06     | 4.41     | 4.11        | 4.01             |
>
> **(2) Hierarchical data generation strategy**
>
> We prompted GPT-4-Turbo-0125 to generate preferences freely (a preference set of four values or a single detailed preference description) and qualitatively compared them to our original hierarchically generated ones. On ten samples, we observed that free-form preference generation can create more topic-specific preferences, but oftentimes they deviated from what we expect preferences to be. Specifically, some generations included preferences irrelevant to the goal of the user instruction (e.g., a preference for nutritional benefits in a *math* problem illustrated with apples) or attempted to resolve the user request and hint solutions (e.g., explicitly instructing correct implementations for a coding problem).
>
> These problems arise because the model needs to understand and elicit preferences from the human user side, not the assistant side. Our hierarchical data generation strategy allows sufficient control over synthesizing *what users would expect in the response*. We have decided that preferences for any response would differ under the style, background knowledge, informativeness, and harmlessness dimensions based on various existing literature (See Appendix A.2). Our method of providing the dimensions in context, coupled with manually crafted seed examples, is instrumental in obtaining high-quality, individualized preferences.
>
> Please see our global response for further verification of our synthetic data and method.
>
> ---
>
> **Effectiveness of our method as model scales (W3, Q3)**
>
> We explored how variations in the type and size of the base model affect performance. Initially, we used Mistral 7B v0.2, but we also experimented with LLaMA models. The results in the table indicate that LLaMA 2 7B performs similarly to or slightly worse than Mistral 7B v0.2. However, increasing the model size from 7B to 13B (LLaMA 2 7B vs. 13B) results in a clear performance improvement. Notably, the latest model, LLaMA-3 8B, surpasses the larger LLaMA 2 13B in benchmark scores (LLaMA 3 8B vs. LLaMA 2 13B). Therefore, both model size and the capabilities of the base pre-trained model significantly impact performance when applying our method.
>
> | Base pre-trained model                  | mf -AlpacaEval | mf-FLASK | mf-Koala | mf-MT-Bench | mf-Self-Instruct |
> | ----------------------- | -------------- | -------- | -------- | ----------- | ---------------- |
> | Mistral 7B v0.2 (→ Janus) | 4.43           | 4.06     | 4.41     | 4.11        | 4.01             |
> | LLaMA 2 7B              | 4.41           | 4.01     | 4.41     | 4.08        | 4.03             |
> | LLaMA 2 13B             | 4.5            | 4.3      | 4.5      | 4.23        | 4.08             |
> | LLaMA 3 8B              | 4.5            | 4.4      | 4.34     | 4.31        | 4.14             |
>
> ---
>
> **Consistency of Janus in maintaining preferences in multi-turn scenarios (Q4)**
>
> To assess whether Janus effectively adheres to diverse preferences in multi-turn scenarios, we conducted an evaluation focusing solely on the multi-turn questions from the MT-Bench, excluding single-turn questions. The results indicate that Janus consistently outperforms the baselines, Mistral 7B Instruct v0.2 and LLaMA-3 8B Instruct (6, 6.6 vs 6.8). This is consistent with the trends observed in Table 3, suggesting that Janus remains highly competitive in multi-turn settings.
>
> | Models | Score [0, 10] |
> | -------------------------------|--------------------|
> | Mistral7B Instruct v0.2 | 6          |
> | LLaMA 3 8B Instruct     | 6.6          |
> | Janus 7B                          | 6.8                |

---

> > ### Comment · Reviewer_eXeU · 2024-08-12
> >
> > Thanks the authors for their response. After reading the response, I think my current score is appropriate.

---

### Author Rebuttal · Authors · 2024-08-07

We provide extra analyses on Multifaceted Collection and Janus in this global response, which we believe will comprehensively address various concerns about our approach. Four aspects of our method are further investigated: quality (G1), diversity (G2), safety (G3), and bias (G4). We attach a **PDF containing three supplementary figures** below.

---

**G1: Quality**

While we have shown the effectiveness of our dataset through the fine-tuned model’s superior benchmark performances (Section 5.1-5.2) and ablation studies (Section 6.1-6.2), we additionally cross-checked the quality of the LLM-generated system messages. Specifically, we developed two criteria that a proper system message should adhere to: (1) relevance and specificity (2) coherence and naturalness. We created a score rubric indicating a scale of 1 to 5 for each criterion.

Inspired by recent works that use LLM-as-a-Judge to assess the process of training data synthesis [1,2], we used LLaMA 3.1 8B Instruct to score a random 20k subset of system message-user instruction pairs. Results show an average of 3.74 on relevance and specificity and 4.01 on coherence and naturalness, with 68.8% and 85.6% instances at or above score 4, respectively. This demonstrates the quality of verbalized preferences, potentially revealing why our model is effectively steerable for pluralistic alignment.

---

**G2: Diversity**

We point to various pieces of evidence presented in our paper regarding the diversity of preferences in the Multifaceted Collection.

- The number of individual preference values embedded in system messages is 797k, derived from 6k subdimensions (see Table 1). Since a single system message is a mixture of preferences from different dimensions, we assert that our dataset contains a wide range of multifaceted human preferences. Example sub-dimensions and keywords of preference descriptions are in Table 7.
- We calculated the ROUGE-L similarities for every possible pair of *preference descriptions* associated with each instruction. The average ROUGE-L score across all dimensions is approximately 0.21, peaking at 0.25 (see Appendix B, Figure 4). Compared to results from previous studies creating synthetic datasets [3], this demonstrates significant diversity among preferences for the same instruction.
- Furthermore, as illustrated in Section 5.1 and Figure 9, we measured the resulting ROUGE-L scores between *responses* generated by language models when different system messages were presented in response to a single user instruction. Janus showed lower ROUGE-L scores compared to the Mistral 7B Instruct v0.2 and GPT-4-Turbo. This also confirms that diversity is learnable from Multifaceted Collection.

In addition, we test if Janus exhibits less similarity to human populations compared to its base pre-trained model (Mistral 7B v0.2) and its post-aligned counterpart (Mistral 7B Instruct v0.2). Following [4], models were evaluated on GlobalOpinionQA and Machine Personality Inventory (MPI), and then we calculated the Jensen-Shannon distance between the human (US and Japan) and model distributions in answer choices. Echoing the findings of [4], Supplementary Figure 1 shows that aligned models including Janus become more distant to the human population after fine-tuning. Still, Janus diverges less from the pre-trained distribution than Mistral 7B Instruct v0.2 does. We also measured the entropy, and the Supplementary Figure 2 visualizes that Janus is significantly closer to pre-trained and human distributions than Mistral 7B Instruct v0.2 does. These experiments suggest that our training method can facilitate calibration to diverse individuals.

---

**G3: Safety**

To check the presence of unsafe content in our synthetic dataset, we evaluated the dataset's system messages, user prompts, and gold responses using a content safety classifier, Llama Guard 3 8B. 99.2% of the 196,998 instances were classified as safe.

Moreover, as presented in Table 4, we have tested Janus on RealToxicityPrompts, showing that Janus has 5.2% and 5.7% lower probability of generating toxic text compared to Mistral 7B Instruct v0.2 and LLaMA 3 Instruct 8B, respectively. This indicates that neither the dataset nor the model is exceptionally unsafe compared to others.

---

**G4: Bias**

Since it is difficult to directly expose biases in our dataset besides the analyses above, we evaluated Janus and three baselines on three social bias benchmarks: Winogender, CrowS-Pairs, and BBQ, all in zero-shot. We include Gemma 2 9B IT as a baseline as it is a SOTA similar-sized model reported to have been extensively tested on bias benchmarks.

| Model | Winogender | CrowS-Pairs | BBQ (Ambig) | BBQ (DisAmbig) |
|-------|------------|-------------|-------------|----------------|
|         | Acc ↑ | Likelihood Diff ↓ / % Stereotype ↓ | Acc ↑ / Bias score ↓ | Acc ↑ / Bias score ↓ |
| Mistral 7B Instruct v0.2 | 0.61 | 4.45 / 67.74 | 0.11 / 11.63 | 0.87 / 2.52 |
| LLaMA 3 8B Instruct | 0.64 | 4.05 / 64.52 | 0.08 / 12.99 | **0.88** / 1.98 |
| Gemma 2 9B IT | **0.68** | 5.44 / **62.43** | **0.42** / **7.62** | 0.86 / **1.33** |
| **Janus 7B** | 0.64 | **4.02** / 67.68 | 0.08 / 12.26 | 0.86 / 3.25 |

According to the results in the table above, Janus shows a degree of bias similar to that of LLaMA 3 8B Instruct across all tasks, and is better than Mistral 7B Instruct v0.2 except in BBQ. Overall, we do not see critical issues of bias in Janus compared to other models, and hypothetically Multifaceted Collection as well. Category-wise Winogender evaluation results are visualized in Supplementary Figure 3.

---

[1] Yuan et al.. Self-Rewarding Language Models. ICML 2024.

[2] Xu et al.. Magpie: Alignment Data Synthesis from Scratch by Prompting Aligned LLMs with Nothing. arXiv 2024.

[3] Honovich et al.. Unnatural Instructions: Tuning Language Models with (Almost) No Human Labor. ACL 2023.

[4] Sorenson et al.. A Roadmap to Pluralistic Alignment. ICML 2024.

---

> ### Author Response · Authors · 2024-08-09
> **Looking forward for Discussions**
>
> Dear Reviewers,
>
> We appreciate for your time and effort on reviewing the paper.
>
> As we near the end of the discussion period, we kindly remind you of the upcoming deadline.
>
> We are eager to discuss any aspects of the paper that may require further clarification.
>
> Thank you once again for your valuable feedback.

---

### Comment · Area_Chair_DS99 · 2024-08-08

Dear reviewers,

The rebuttal phase has now ended and the authors have submitted their responses to the reviews. In the coming days (August 7-13) there will be an open discussion between the reviewers and authors. Please read the responses, respond to them early on in the discussion, and discuss points of disagreement.

Best,

AC

---

### Decision · Program_Chairs · 2024-09-25

**Decision:**

Accept (poster)

**Comment:**

The paper proposes a novel framework for aligning LLMs with individual user preferences through specifying values in system messages. It establishes the Multifaceted Collection dataset and trains the 7B LLM Janus. Janus can generalize to unseen system messages and outperforms models like GPT-3.5 and GPT-4 in adhering to user preferences. It also demonstrates effectiveness in benchmarks that assess response helpfulness, highlighting the potential of this approach for scalable LLM alignment.

Reviewers agree that the paper possesses several strengths. The utilization of diverse system messages for alignment is novel and creative, offering a scalable solution for personalized alignment. The methodology is reasonable, encompassing careful dataset construction and comprehensive evaluations against strong baselines. The paper is well-structured and clearly written.

To further improve the technical quality of the paper, reviewers provide some valuable suggestions. Firstly, the paper should explore potential risks and handle conflicting or ethically questionable preferences. Secondly, the paper should provide the details of human annotation and discuss the role of system messages for individualized alignment. Thirdly, the approach should be tested on models of different sizes to explore scale effects. Fourthly, the authors should include quantitative and qualitative analyses to investigate potential biases in the dataset. Some of the concerns have been addressed in the user rebuttal.